# Delayed short-term tamoxifen treatment does not promote remyelination or neuron sparing after spinal cord injury

**Nicole Pukos**[1,2], **Dana M. McTigue**[2,3] *

**1** Neuroscience Graduate Program, The Ohio State University, Columbus, OH, United States of America,
**2** Belford Center for Spinal Cord Injury, The Ohio State University, Columbus, OH, United States of America,
**3** Department of Neuroscience, Wexner Medical Center, Ohio State University, Columbus, OH, United States of America

* Dana.McTigue@osumc.edu

**Data Availability Statement:** All data files have been uploaded to Open Data Commons for Spinal Cord Injury database, DOI: 10.34945/F5QP4H

**Funding:** The grants NS100522 and P30 NS104177 were awarded to DMM from the

## Abstract

The tamoxifen-dependent Cre/lox system in transgenic mice has become an important research tool across all scientific disciplines for manipulating gene expression in specific cell types. In these mouse models, Cre-recombination is not induced until tamoxifen is administered, which allows researchers to have temporal control of genetic modifications. Interestingly, tamoxifen has been identified as a potential therapy for spinal cord injury (SCI) and traumatic brain injury patients due to its neuroprotective properties. It is also reparative in that it stimulates oligodendrocyte differentiation and remyelination after toxin-induced demyelination. However, it is unknown whether tamoxifen is neuroprotective and neuroreparative when administration is delayed after SCI. To properly interpret data from transgenic mice in which tamoxifen treatment is delayed after SCI, it is necessary to identify the effects of tamoxifen alone on anatomical and functional recovery. In this study, female and male mice received a moderate mid-thoracic spinal cord contusion. Mice were then gavaged with corn oil or a high dose of tamoxifen from 19–22 days post-injury, and sacrificed 42 days post-injury. All mice underwent behavioral testing for the duration of the study, which revealed that tamoxifen treatment did not impact hindlimb motor recovery. Similarly, histological analyses revealed that tamoxifen had no effect on white matter sparing, total axon number, axon sprouting, glial reactivity, cell proliferation, oligodendrocyte number, or myelination, but tamoxifen did decrease the number of neurons in the dorsal and ventral horn. Semi-thin sections confirmed that axon demyelination and remyelination were unaffected by tamoxifen. Sex-specific responses to tamoxifen were also assessed, and there were no significant differences between female and male mice. These data suggest that delayed tamoxifen administration after SCI does not change functional recovery or improve tissue sparing in female or male mice.

National Institute of Neurological Disorders and Stroke (https://www.ninds.nih.gov/) and funded this research. The funder had no role in study design, data collection and analysis, decision to publish, or preparation of the manuscript.

**Competing interests:** The authors have declared that no competing interests exist.

## Introduction

Spinal cord injury (SCI) is a traumatic life-changing event that currently affects ~300,000 people in the United States [1]. The clinical signs associated with SCI depend on the level and severity of injury, but often include partial or complete loss of sensory, motor, and autonomic control below the level of injury. These deficits result from tissue damage incurred by the initial impact of injury, and are exacerbated by secondary pathophysiological mechanisms. Among these are inflammation [2–6], axon degeneration [7–10], glial reactivity [11, 12], and demyelination [13–16], which spread above and below the injury site and can persist for months post-injury [for review see [17]. Due to their contributions to pathology, these events are often therapeutic targets for SCI research.

One common approach for studying secondary injury mechanisms after SCI is using transgenic mouse models because of their utility in targeting specific genes. A major advantage of transgenic mice is the Cre/lox recombination system, which revolutionized the scientific community by providing a tool to control gene activity in most mouse tissues. In the case of SCI, these sophisticated mouse models are used to study the mechanisms that contribute to widespread tissue damage and to identify new therapeutic targets. The Cre/lox system is based on Cre recombinase, a P1 bacteriophage that cuts out DNA sequences between loxP sites in any cell with active Cre [18]. However, since some phenotypes are embryonically lethal, traditional Cre mouse models cannot always be used. To optimize gene targeting and to control the timing or location of Cre activity, a ligand-dependent regulatable Cre recombinase was developed, called CreER [19, 20]. CreERT2 is the most effective CreER [21] and consists of Cre bound to a mutated estrogen receptor that has a high affinity for tamoxifen and a low affinity for estrogen [21]. After tamoxifen binds to the mutated estrogen receptor, CreERT2 translocates from the cytoplasm to the nucleus where it excises the floxed DNA. By combining tamoxifen-dependent Cre recombinase activity with tissue-specific expression of CreERT2, gene expression can be controlled temporally and spatially.

Tamoxifen is a selective estrogen receptor modulator that acts as an agonist or antagonist on the estrogen receptor in a tissue-specific manner. The action of tamoxifen depends on the estrogen receptor complex and ligand-binding affinity [22–25]. For example, tamoxifen is an estrogen receptor agonist in the bone and uterus [23, 26], but an antagonist in the breast [22, 27, 28]. In the breast, estrogen stimulates mammary cell proliferation, thereby increasing the chances of DNA mutations that may lead to breast cancer [29]. Because tamoxifen acts as an antagonist to the estrogen receptor in mammary cells, it reduces proliferation and slows tumor growth [30, 31], making it an effective therapeutic for breast cancer patients. In addition to its peripheral effects, tamoxifen crosses the blood-brain barrier [32] and has shown neuroprotective effects in models of stroke [33], dementia [34], multiple sclerosis [35], Parkinson's disease [36], traumatic brain injury [37], and SCI [38]. For example, tamoxifen injected intraperitoneally in adult female rats acutely after toxin-induced demyelination in the brain stimulated NG2 + oligodendrocyte progenitor cell differentiation into oligodendrocytes [OLs] and accelerated the rate of remyelination [35]. Furthermore, 21 days of tamoxifen treatment beginning immediately after SCI reduced inflammation, decreased glial reactivity, promoted cell survival, increased white matter sparing, and enhanced functional recovery in rats and cats [38–43]. Interestingly, if tamoxifen treatment was delayed until 42 days after SCI, then hindlimb motor recovery was not changed; the impact of delayed tamoxifen treatment on tissue repair was not assessed [43].

The neuroprotective effects of tamoxifen have been extensively studied in CNS disease and traumatic injury models. However, the impact of short-term tamoxifen treatment for the purpose of Cre recombination has yet to be determined after SCI. Ideally, controls for genetic

studies would include a knock-out and wild-type strain that both receive tamoxifen, but this is not always possible. For example, reporter mouse studies that use GFP to track cell fate over time typically have a STOP codon before the GFP promoter. After tamoxifen is given, the floxed STOP codon is removed which drives GFP expression in Cre expressing cells. Without tamoxifen, it would be impossible to induce cell-specific GFP expression in controls. In addition, previous studies that propose tamoxifen as a neuroprotective treatment have given tamoxifen within 3 days of disease or trauma [35, 38–42]. It is unknown if delayed tamoxifen treatment yields similar results. After SCI, remyelination does not begin until 2–3 weeks post-injury, therefore delayed tamoxifen treatment is required when using Cre mice to manipulate gene expression during this time frame [44]. To determine if delayed tamoxifen changes motor recovery or tissue sparing after SCI, we treated 3 week post-injury mice with tamoxifen and then examined myelination, OL number and NG2+ progenitor cell proliferation, astrocyte and macrophage reactivity, neuron and axon number, axon sprouting, and hindlimb motor recovery. Sex differences after tamoxifen administration were also examined. Tamoxifen was given via oral gavage for 4 consecutive days; this dosing regimen is sufficient to induce Cre recombination in transgenic mice [44], and thus is widely used across scientific disciplines. The data reveal that delayed short-term tamoxifen treatment is not neuroprotective and does not alter motor recovery in male or female mice after SCI. These data are applicable to Cre mouse models and suggest that behavioral and histological changes observed following tamoxifen induced Cre-mediated recombination, at least in the fashion in which we gave it, are not due to tamoxifen treatment.

## Methods

### Animals

All surgical and postoperative care procedures in this study were approved and performed in accordance with the Ohio State University Institutional Animal Care and Use Committee (Protocol Number: 2012A00000030-R2). Twelve-week-old female and male C57BL/6J (WT) mice were purchased from The Jackson Laboratory and group housed under traditional conditions: a 12 hour light-dark cycle with ad libitum access to food and water. Animals did not receive analgesics during the study, as approved in our protocol; however, all efforts were made to minimize animal suffering for the duration of the experiment. Alpha Dri bedding (Cincinnati Lab Supply Inc.) was added to cages to provide additional comfort, and mice were assessed twice daily for discomfort; no infections, wounds, or distress were recognized during the study.

### Spinal cord injury

Mice were anesthetized with a ketamine/xylazine mixture (80 and 10mg/kg, respectively) and received a single-level laminectomy at the vertebral level T9. The animals then received a moderate (75 kDyne force) spinal cord contusion using the Infinite Horizons device (Precision Instruments) (mean force: 77.15 kDyne, mean displacement: 600.30 μm). The peak displacement for one male mouse and one female mouse was greater than two standard deviations from the mean, so these mice were considered outliers and excluded from the study. Muscle overlaying the spinal cord was closed with dissolvable sutures, and the skin was closed with surgical clips. Mice were kept on warmers (32˚C) overnight. Postsurgical care included 5-day administration of antibiotics (gentomicin, 5mg/kg) and saline subcutaneously for hydration. Bladders were manually expressed twice per day for the duration of the study. One male mouse from each group died before sacrifice for unknown reasons.

## Tamoxifen administration

At 19 days post-injury, mice were randomly divided into tamoxifen or control groups. Tamoxifen (Sigma) was dissolved in corn oil (Spectrum) (40 mg/kg) by overnight shaking at 37˚C. Tamoxifen was administered by oral gavage (300 mg/kg) to spinal cord injured mice (n = 9 male; n = 10 female) from 19–22 days post-injury. Control spinal cord injured mice received corn oil gavage (n = 8 male; n = 9 female).

## 5-ethynyl-2'-deoxyuridine labeling and detection

5-ethynyl-2'-deoxyuridine (EdU) (Invitrogen) is a thymidine analogue that is incorporated into replicating DNA and used to detect proliferating cells. EdU was dissolved in the drinking water (0.2 mg/ml) and administered to mice from 21–42 days post-SCI via light-protected water bottles. EdU at this dose is nontoxic [45]. EdU water was available ad libitum and replaced every other day. EdU labeling was detected in tissue sections using the AlexaFluor-647 Click-iT EdU Cell Proliferation Kit for Imaging (Invitrogen) immediately following the final step of immunofluorescent staining. The protocol was followed according to the manufacturer's instructions. Briefly, 10 μm longitudinal (horizontal) spinal cord sections were washed in 0.5% Triton X-100/PBS, incubated in the Click-iT reaction cocktail for 30 min at room temperature in the dark, rinsed with PBS, and then coverslipped with Immu-Mount (Thermo Scientific).

## Behavioral analysis

Hindlimb motor recovery was assessed using the Basso Mouse Scale [46] and automated horizontal ladder. Mice were acclimated to the open field and assessed before injury for normal walking (score of 9) and tested at 1, 7, 14, 21, 28, 35, and 42 days post-injury (dpi). Two raters blind to study groups scored locomotor performance over 4 min considering ankle movement, weight support, plantar stepping, coordination, paw position, and trunk stability. Scores range from no hindlimb movement (0) to normal locomotion (9) and were averaged for right and left hindlimb. Once mice regained plantar stepping ability (BMS score of 4), they were tested on the automated horizontal ladder (Columbus Instruments, Ohio) to test precision and coordination of stepping. The ladder has evenly spaced metal rungs 0.85 cm apart for a total distance of 86.25 cm. Mice walked the full length of the ladder and missed steps were automatically registered by a pan below the rungs. Passes where mice turned around or stopped before the end of the ladder were discounted. Total missed steps represent the average of three trials at each time point. Locomotor function in both behavior tests was normal before injury.

Gross motor activity was further assessed using the activity box (AccuScan Instruments Incorporated). Baseline data was collected and mice were tested at 1, 8, 15, 22, 29, 36, and 41dpi. Horizontal and vertical beams detected ambulation, rearing, and velocity which were tracked using Fusion 6.4 software and recorded over 10 min.

## Tissue processing

A subset of mice were euthanized 42dpi (corn oil female n = 6; tamoxifen female n = 7; corn oil male n = 5; tamoxifen male n = 6) with a lethal dose of ketamine/xylazine mixture (1.5X surgical dose) and perfused transcardially with 0.1M PBS, followed by 4% paraformaldehyde (PFA) in PBS (10ml/min flow rate). Spinal cords were removed, post-fixed in 4% PFA for 2 h, then transferred to 0.2 M PB overnight. The next day, tissue was cryoprotected for 3 days in 30% sucrose dissolved in distilled water. 1 cm segments of spinal cords centered on the lesion

were embedded in optimum current temperature (OCT) compound (Electron Microscopy Sciences) and frozen on dry ice. Blocked spinal cords were cut in serial longitudinal sections (ventral to dorsal) at 10 μm on a cryostat, then tissue was mounted on SuperFrost Plus slides (Thermo Fisher Scientific) and stored at -20°C until use.

The remaining mice from each cohort were euthanized at 42dpi (n = 3 per group) for spinal cord Epon embedding. Spinal cord injured mice were exsanguinated with 0.1M PBS then perfused with 4% PFA/2% glutaraldehyde in 0.1M PBS. 5 mm sections centered on the lesion were dissected from the spinal cord and fixed with 4% PFA/2% glutaraldehyde overnight. Sections were further cut into 1 mm segments containing the lesion, and segments rostral and caudal to the lesion. Spinal cord segments were then processed for Epon embedding as described previously [47, 48]. Semi-thin sections were cut longitudinally at 1 μm on an ultramicrotome (Ultracut UCT; Lecia). Sections were stained for myelin using 1% toluidine blue/ 1% borax, dehydrated through a graded ethanol series, and coverslipped.

## Immunohistochemistry

Sections were rinsed in 0.1M PBS, then endogenous peroxidase activity was blocked using 10:1 methanol/30% hydrogen peroxide for 20 min in the dark. After rinsing with PBS, sections were blocked for nonspecific antigen binding using 4% BSA/0.3% Triton X-100/PBS (BP$^{3+}$) for 1 h in a humid chamber. Primary antibodies (Table 1) were diluted in BP$^{3+}$ and applied to sections overnight at room temperature. Next, sections were rinsed and treated with biotinylated secondary antibodies for 1 h at room temperature. Sections were primed with Elite avidin-biotin enzyme complex (ABC; Vector Laboratories) for 1 h, and labeling was visualized with ImmPACT$^{TM}$ DAB (Vector Laboratories). Sections were rinsed, dehydrated, and coverslipped with Permount (Thermo Fisher Scientific).

## Immunofluorescence

Sections were rinsed in 0.1M PBS and blocked for nonspecific antigen binding using 4% BSA/ 0.1% Triton X-100/PBS (BP$^{+}$) for 1 h in a humid chamber. Primary antibodies (Table 1) were diluted in BP$^{+}$ and sections were incubated overnight at room temperature. Following PBS rinses, Alexa Fluor secondary antibodies (1:1000; Invitrogen) were applied for 1 h in the dark. Sections were rinsed and incubated in DAPI (1:50,000; Invitrogen) for 15 min in the dark to label nuclei. After rinses, slides were coverslipped with Immu-Mount (Thermo Scientific). Since node of Ranvier staining involves two rabbit primary antibodies, a modified protocol

**Table 1. List of primary antibodies used in this study.**

| Primary antibody (IHC/IF): specificity | Concentration | Host species | Vendor | Catalogue number |
|---|---|---|---|---|
| NeuN (IHC): neurons | 1:10,000 | Rabbit | Abcam | ab104225 |
| GAP-43 (IHC): axon sprouting | 1:5000 | Rabbit | Fisher | AB5220MI |
| NF-H (IHC & IF): large diameter axons | 1:1000 | Chicken | Aves Labs | Aves |
| GSTπ (IF): oligodendrocytes | 1:1000 | Rabbit | Biorbyt | orb18037 |
| CD68 (IF): activated macrophages/microglia | 1:1000 | Rat | BioRad | MCA1957GA |
| GFAP (IF): astrocytes | 1:1000 | Chicken | Aves Labs | Aves |
| NG2 (IF): progenitor cells | 1:1000 | Rabbit | Millipore | AB5320 |
| Kv1.2 (IF): juxtaparanode of Ranvier | 1:1000 | Rabbit | Abcam | APC-010 |
| Caspr (IF): paranode of Ranvier | 1:1000 | Rabbit | Abcam | ab34151 |

Primary antibodies are identified based on target and use in immunohistochemistry (IHC) and/or immunofluorescence (IF), concentration, host species, vendor, and catalogue number.

was followed. Sections were rinsed in 0.1M PBS and blocked for nonspecific antigen binding using BP$^{3+}$ for 1 h in a humid chamber. Sections were incubated in anti-Caspr antibody (1:1000; Abcam) diluted in 4% BSA/0.3% TritoX-100/3% normal goat serum/PBS (BPG$^{3+}$) for 3 h, rinsed with PBS, then incubated in Alexa Fluor 546 goat anti-rabbit secondary antibody (1:1000; Invitrogen) diluted in BPG$^{3+}$ for 1 h at room temperature. After rinsing with PBS, heat-mediated antigen retrieval was performed. Sections were placed in antigen retrieval solution containing 0.1% high pH solution in dH$_2$O heated to 90°C for 10 min, then cooled at room temperature for 10 min. Sections were washed with PBS, blocked in BPG$^{3+}$ for 1 h, then incubated in anti-Kv1.2 antibody (1:1000; Alomone) diluted in BP$^+$ overnight at room temperature. Sections were again rinsed with PBS and treated with Alexa Fluor 488 goat anti-rabbit secondary antibody (1:1000; Invitrogen) diluted in BP$^{3+}$ for 1 h. After washing sections with PBS, they were coverslipped with Immu-Mount (Thermo Scientific). Sections labeled with GST$\pi$ also underwent antigen retrieval using high pH to improve antibody binding. All protocols have been verified with no-primary and no-secondary controls.

## White matter sparing

Lesion area and spared white matter were assessed by staining myelin with eriochrome cyanine (EC), as previously described [49]. The lesion epicenter was defined as the area with the least intact myelin (EC+ stain) and the largest central core lesion. Images were taken of longitudinal spinal cord sections at the level of the central canal using a Zeiss Axiocam 305 color microscope at 5x magnification. Three images spanning 3 mm total were taken for each animal, including an image centered on the lesion and evenly spaced sections 1 mm rostral and 1 mm caudal from the lesion epicenter. Total white matter sparing was measured using Materials Image Processing and Automated Reconstruction software (MIPAR™) to identify EC-positive staining (target area), and automatically calculated based on total spinal cord area (proportional area = target area/total spinal cord area). Only compact EC surrounding the lesion and in white matter rostral and caudal to the lesion was included in the area of spared white matter calculation.

## Axon counts

A total of twelve images per animal were taken at 40x (Zeiss Axiocam 305 color microscope). Images were taken on the left and right side of the central canal in the lateral white matter at 2 mm, 1 mm, and 0.5 mm rostral and caudal to the lesion (Fig 1). Thresholding for NF-H and GAP-43 immunoreactivity and dividing by total section area produced axon proportional area and axon sprouting proportional area, respectively. Areas from left and right white matter images were averaged at each distance for a total of 6 values per section.

## Neuronal survival quantification

Longitudinal spinal cord sections of the dorsal and ventral horn were imaged 1 mm rostral and caudal to the lesion using a Zeiss Axiocam 305 color microscope at 5x with a 1.6 zoom. Neuron counts were quantified using MIPAR by thresholding for NeuN labeling and dividing by total gray matter area.

## Immunofluorescence microscopy and quantification

Confocal z-stacks with a 1.0 μm step-size spanning 10 μm were collected at 2 mm, 1 mm, and 0.5 mm rostral and caudal to the lesion at the level of the central canal (see Fig 1) using a Leica SP8 confocal microscopy system at 40x magnification. Z-stacks for each spinal cord section

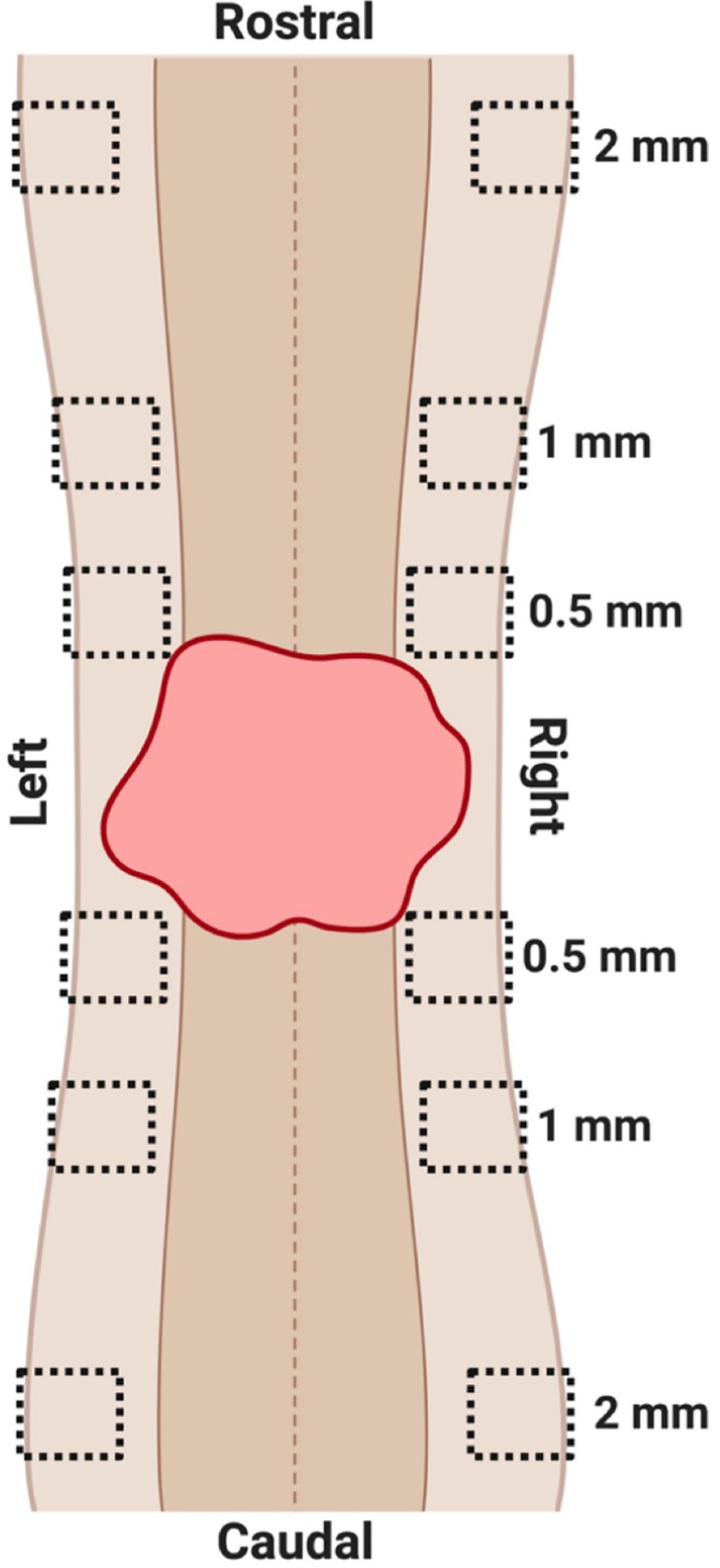

**Fig 1. Diagram of longitudinal spinal cord 2 mm rostral and caudal to the lesion epicenter demonstrating sample box placement during imaging.** The central canal is depicted as a dotted line running through the center of the cord; lesioned tissue is the red irregular shape; gray matter is the dark brown on either side of the center canal; white matter is the light brown on either side of the gray matter. Sample boxes were placed in the right and left white matter, with one box at 2 mm, 1 mm, and 0.5 mm rostral and caudal to the lesion epicenter.

were compressed into one maximum intensity projection using Leica Application Suite X software. Values from left and right white matter images were averaged at each distance for a total of 6 counts per section.

**Node of Ranvier quantification.** Intact nodes of Ranvier, classified as Kv1.2 profiles flanking two Caspr profiles, were counted using MIPAR Deep Learning and Image Processor. First, 5 maximum intensity projection confocal images were manually traced using MIPAR to identify nodes of Ranvier. These 5 traces, along with the 5 original images, were then uploaded to the MIPAR Deep Learning Trainer. After training finished, the generated model was further refined with additional post-processing steps using the MIPAR Image Processor. This final recipe was used to count nodes of Ranvier and hemi-nodes (defined as one Kv1.2 segment adjacent to a Caspr segment) for all remaining images. The percent of intact nodes of Ranvier was calculated by dividing the total number of nodes by the number of hemi-nodes, and multiplying by 100. Nodes of Ranvier per axon area was calculated by dividing the number of intact nodes of Ranvier by axon proportional area.

**Proliferating cell count.** EdU+ cells, proliferating NG2 cells (NG2/EdU), and proliferating CNS macrophages (CD68/EdU) were manually counted for each section. Proliferating cells were counted if EdU labeling co-localized with DAPI. Double labeled cells were counted if a well-defined border of NG2 or CD68 labeling surrounded an EdU+ nuclei.

**Oligodendrocyte cell counts.** GST$\pi$+ labeling co-localized with DAPI was analyzed using MIPAR. Briefly, a MIPAR recipe first thresholded for DAPI and GST$\pi$+ labeling, then quantified the total number of overlapping events.

## Astrocyte reactivity quantification

Tiled confocal z-stacks centered on the lesion were acquired at 20x magnification (Leica SP8 confocal microscope). Lesion size and astrocyte reactivity were analyzed using customized recipes created in MIPAR. Injury cavities were identified as GFAP negative regions surrounded by dense GFAP positive labeling. Lesion borders were outlined to measure the area of the lesion. GFAP area was calculated by thresholding for GFAP immunoreactivity and dividing by total section area.

## Statistical analysis

All data were collected and analyzed in a blinded manner. Data were analyzed using GraphPad software (Prism 8) and are presented as mean + SEM, with group differences reaching statistical significance at $p < 0.05$. Behavioral data, white matter sparing, axon counts, and neuron counts were analyzed by repeated measures two-way ANOVA followed by the Šídák *post hoc* test. Astrocyte reactivity, proliferating cell counts, OL counts, node of Ranvier counts, and axon myelination were analyzed by a mixed-effect model followed by the Šídák *post hoc* test because missing values that resulted from wrinkled or missing sections. All histological plates were assembled using Adobe Photoshop CC 20.0.4; when necessary, contrast was adjusted equally across all images within a plate to reproduce images as originally viewed through the microscope.

## Results

### Delayed short-term tamoxifen treatment does not impact functional recovery

Locomotor function was assessed 1 week before SCI, 1dpi, and weekly thereafter via BMS, the automated horizontal ladder, and activity box (Fig 2A). All mice displayed normal ambulation in the open field before injury, indicated as a 9 on the BMS scale. As expected with a moderate contusion injury, locomotor function dropped and then spontaneously recovered to a plateau at 14dpi. Note, at this point mice had not received tamoxifen. By 14dpi, all female and male mice achieved frequent or consistent plantar stepping with no coordination, indicated as a 5 on the BMS scale. Corn oil or tamoxifen gavage from 19-22dpi did not alter subsequent BMS scores, which were maintained at a 5. Coordination on the BMS scale is assessed when a

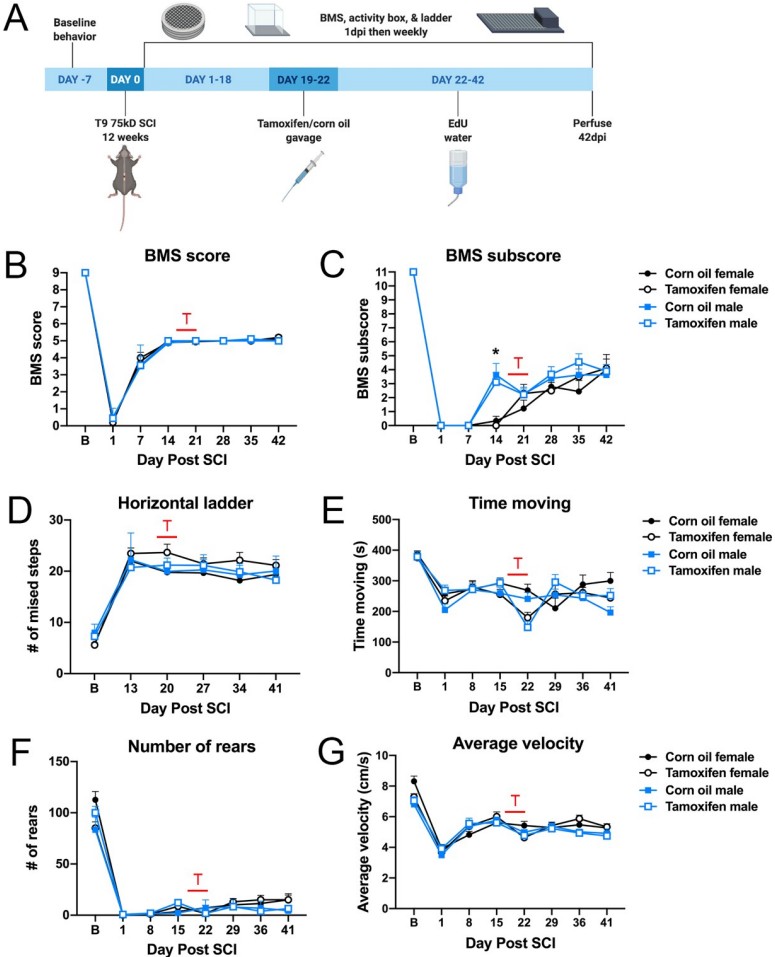

**Fig 2. Motor recovery is unaffected by delayed tamoxifen administration.** (A) Schematic of experimental timeline. (B-G) T indicates when animals received tamoxifen or corn oil gavage. (B) BMS scores were not different between sexes or tamoxifen and control mice at any time post-injury. (C) BMS subscores were significantly increased 14dpi in male mice compared to females, but sex did not impact subscores after gavage. BMS subscores were unaffected by tamoxifen. (D) The number of missed steps on the automated horizontal ladder was not statistically significant between groups. (E) Tamoxifen gavage reduced the number of ambulatory activities at 22dpi in female and male mice. Ambulation of tamoxifen mice recovered by 29dpi. (F-G) (F) The number of vertical rears or (G) speed of ambulation was not different between groups over time. (B-F) Two-way repeated measures ANOVA with Šídák *post hoc* test; corn oil female n = 9; tamoxifen female n = 10; corn oil male n = 8; tamoxifen male n = 9; *p<0.05.

mouse has at least three accessible passes, defined as moving at a consistent speed for at least three body lengths [46]. By 42dpi, 11% of control female mice and 20% of tamoxifen female mice had achieved some forelimb-hindlimb coordination, indicated as a 6 on the BMS scale, but this was not statistically significant (Fig 2B). Male mice in both groups achieved forelimb-hindlimb coordination, but it was not scored since the mice did not have three accessible passes. BMS scores were not statistically different for drug administration or sex (Fig 2B). BMS subscores were calculated to assess fine changes in motor recovery. At 14dpi, male mice scored significantly higher on the BMS subscore scale, averaging a score of ~3.5 compared to an average of 0 for female mice. This increased subscore is attributed to consistent plantar stepping and parallel paw position at initial contact and lift off, and suggests male mice had more fine motor skills before tamoxifen administration. However, during corn oil or tamoxifen gavage, sex no longer significantly impacted motor recovery. 44% of control and 60% of tamoxifen female mice, and 63% of control and 44% of tamoxifen male mice demonstrated consistent plantar stepping. By 42dpi, 100% of control and 80% of tamoxifen female mice, and 88% of control and 67% of tamoxifen male mice achieved consistent plantar stepping and displayed mild trunk stability. Overall, BMS subscores were significantly increased in male mice 14dpi, but unaffected by tamoxifen administration at any time post-injury (Fig 2C).

To further test hindlimb coordination and precision, mice were tested on the automated horizontal ladder. Baseline data revealed a limited number of foot falls for all non-injured mice. By 13dpi, all mice were frequently stepping during BMS and therefore tested on the ladder. All control and treated female and male mice averaged 21–23 missed steps for the duration of the study (Fig 2D), mirroring the plateau in motor recovery observed with BMS (Fig 2B).

Previous studies report lethargy as a side effect of tamoxifen [50, 51]. Our study confirms these findings as indicated by a 1.5-fold decrease in time spent moving during tamoxifen gavage, although the data are not significant (Fig 2E). Once tamoxifen gavage ceased, activity increased and mice no longer appeared lethargic (Fig 2E). We also quantified the total number of vertical rears (Fig 2F) and the mean average velocity for each ambulation (Fig 2G). Neither measure was significantly different for either group. Collectively, these data show that tamoxifen gavage 19-22dpi does not alter locomotor recovery.

## Tamoxifen is not neuroprotective when gavaged 21d post SCI

As a consequence of primary and secondary injuries, myelin, neuron, and axon loss spreads rostral and caudal from the lesion epicenter and contributes to functional deficits [52–54]. Studies have shown that tamoxifen administration immediately after SCI reduces this response in part by promoting cell survival [39, 41, 55]. The effects of delayed tamoxifen administration, however, are not known. To characterize potential neuroprotective effects of delayed tamoxifen treatment, we analyzed white matter sparing, neuron survival, axonal sprouting, and axon number above and below the lesion epicenter.

Longitudinal spinal cord sections were stained with eriochrome cyanine (EC) to label myelin. As expected, the lesion epicenter had the least amount of EC staining with ~11% stained area, which increased 3-fold 1 mm rostral and caudal to the lesion (Fig 3A and 3B). There were no significant differences in the spared white matter between any groups (Fig 3B).

Neurons were quantified in the gray matter of the dorsal and ventral horn just distal to the lesion epicenter. Tamoxifen-treated mice had significantly fewer neurons in the dorsal horn (two-way repeated measures ANOVA; $p = 0.0438$) and ventral horn ($p = 0.0431$) compared to control mice (Fig 4A–4D). However, *post hoc* analysis did not show significant differences in the number of NeuN+ cells rostral or caudal to the lesion epicenter within specific subregions

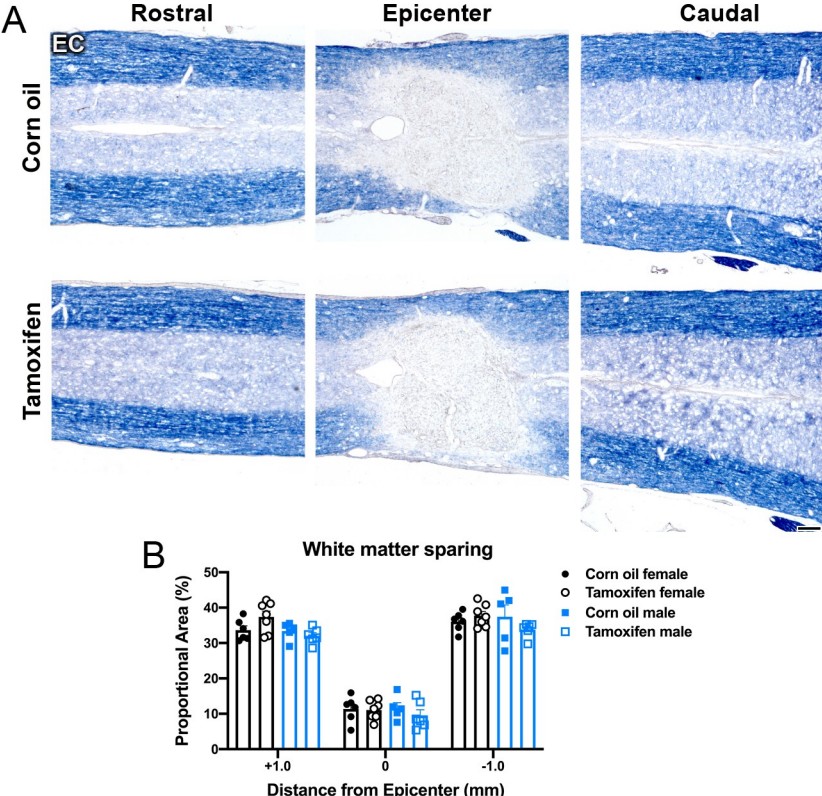

**Fig 3. Delayed tamoxifen treatment does not change the amount of white matter spared at the lesion epicenter, or distal to the lesion.** (A) Representative images of control and tamoxifen mice stained for eriochrome cyanine (EC) at the lesion epicenter, and 1 mm rostral and caudal to the epicenter. Scale bar = 100 μm. (B) Area of EC staining was not significantly different between control and tamoxifen mice or female and male mice at 1 mm rostral to the epicenter, at the epicenter, or 1 mm caudal to the epicenter. Data presented as mean + SEM. Two-way repeated measures ANOVA with Šídák *post hoc* test; corn oil female n = 6; tamoxifen female n = 7; corn oil male n = 5; tamoxifen male n = 6.

(Fig 4A–4D). Sex had no effect on neuron number in the dorsal or ventral horn (Fig 4C and 4D).

We next quantified axonal sprouting near the lesion border and distal to the lesion. Growth associated protein-43 (GAP-43)+ immunolabeling was highest closest to the lesion epicenter in both tamoxifen and control mice, and then gradually decreased with increasing distance from the lesion (Fig 5A and 5C). Neither tamoxifen nor sex had an effect on axonal sprouting. There was also no change the overall area of large caliber axons. At 0.5 mm and 1 mm rostral and caudal to the lesion, neurofilament heavy (NF-H) occupied ~4.5% and ~5.5% for control and tamoxifen female mice, respectively (Fig 5B–5D). 2 mm rostral to the lesion, control female mice remained at 4.5% NF-H+ area, while tamoxifen female mice increased slightly to 6.9% NF-H+ area (Fig 5B–5D), yet this difference was not significant. At 2 mm caudal to lesion, NF-H+ area rose to 7.3% and 7.9% for control and tamoxifen mice, respectively (Fig 5B–5D). In male mice, ~4.5% of control and ~4.1% of tamoxifen mice had NF-H labeling in the lateral white matter at all distances from the lesion epicenter (Fig 5B–5D). Overall, there were no significant differences between groups at any distance. Taken together, these data indicate that delayed 4-day tamoxifen gavage does not increase white matter sparing, neuron survival, axon sprouting, or axon number.

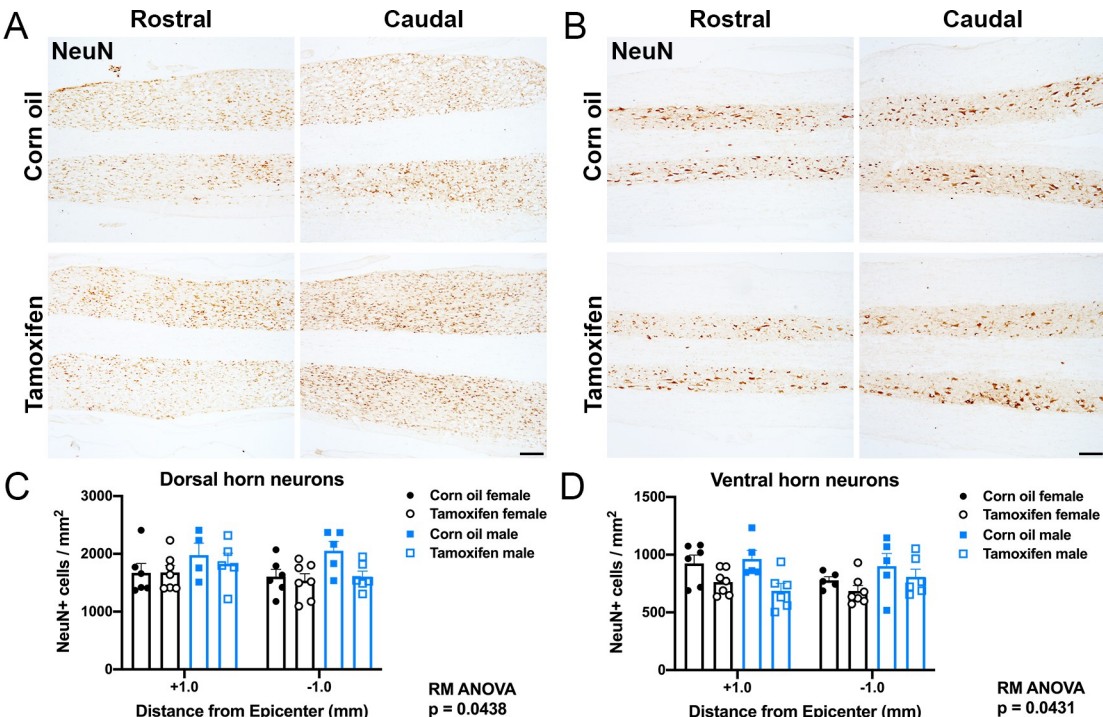

**Fig 4. Neuronal survival after SCI does not change with delayed tamoxifen treatment.** (A-B) Representative images of NeuN labeling in the gray matter of the (A) dorsal horn and (B) ventral horn 1 mm rostral and caudal to the lesion epicenter after corn oil or tamoxifen gavage. Scale bar = 100 μm. (C-D) Tamoxifen mice had significantly fewer NeuN+ cells in the (C) dorsal horn (p = 0.0438) and the (D) ventral horn overall compared to control mice (p = 0.0431). Sex did not impact neuron number in the dorsal or ventral horn. Data presented as mean + SEM. Mixed-effects analysis with Šídák *post hoc* test; corn oil female n = 6; tamoxifen female n = 7; corn oil male n = 5; tamoxifen male n = 6.

## Delayed tamoxifen treatment does not alter astrogliosis or lesion size

Tamoxifen is a known repressor of astrocyte reactivity acutely after SCI [42, 43, 55]. Here, we labeled for GFAP to determine if tamoxifen delivered after glial scar formation yielded similar results. The data reveal that tamoxifen treatment from 19-22dpi did not change the lesion size (Fig 6A–6C) or proportional area (Fig 6A, 6B and 6D) of GFAP+ cells surrounding the lesion in female or male mice.

## Phagocytic monocyte and NG2 cell proliferation are not affected by delayed short-term tamoxifen gavage after SCI

In this study, EdU was given after tamoxifen treatment to evaluate potential tamoxifen effects on cell proliferation at varying distances from the lesion epicenter. EdU+ cells were highest close to the lesion for all groups (Fig 7A and 7B). Control female mice averaged 106 ± 17 EdU + cells 0.5 mm rostral and 124 ± 20 EdU+ cells 0.5 mm caudal to the lesion, and tamoxifen female mice averaged 150 ± 17 EdU+ cells 0.5 mm rostral and 161 ± 12 EdU+ cells 0.5 mm caudal to the lesion (Fig 7A and 7B). A similar trend occurred in male mice, with control mice averaging 159 ± 127 EdU+ cells 0.5 mm rostral and 119 ± 25 EdU+ cells 0.5 mm caudal to the lesion, and tamoxifen mice averaging 201 ± 22 EdU+ cells 0.5 mm rostral and 198 ± 34 EdU + cells 0.5 mm caudal to the lesion (Fig 7B). In all groups, the number of proliferating cells gradually decreased as the distance from the epicenter increased (Fig 7A and 7B). No significant differences were detected between control and tamoxifen groups.

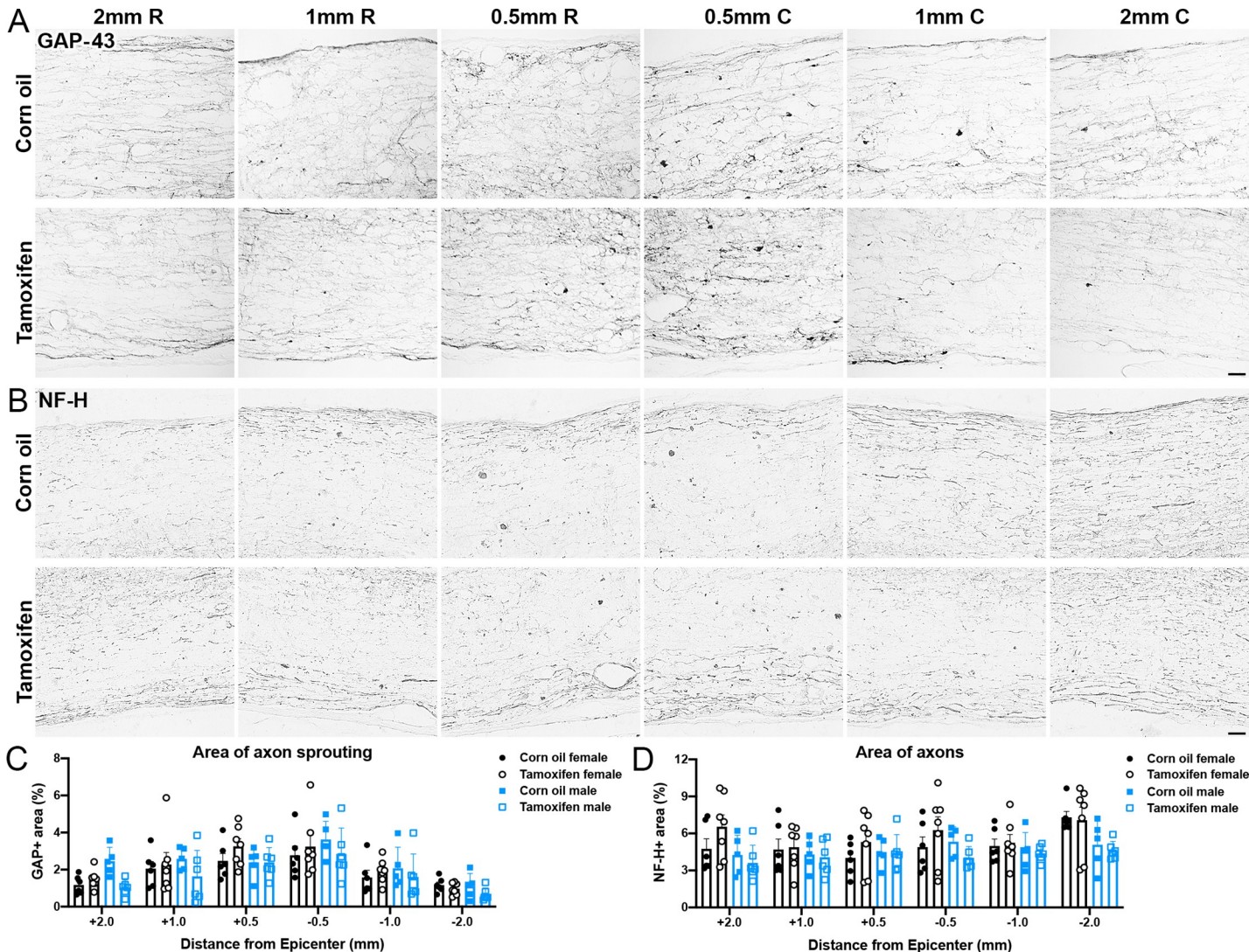

**Fig 5. Delayed tamoxifen treatment does not alter total axon area after SCI.** (A-B) Representative images of longitudinal spinal cord sections immunolabeled for (A) GAP-43 and (B) neurofilament-heavy (NF-H). Scale bar = 20 μm. (C-D) Percent area of (C) GAP-43 and (D) NF-H+ axons in the lateral white matter 2 mm, 1 mm, and 0.5 mm rostral and caudal to the lesion epicenter. No significant differences are reported between groups. Data presented as mean + SEM. Two-way repeated measures ANOVA with Šídák *post hoc* test; corn oil female n = 6; tamoxifen female n = 7; corn oil male n = 5; tamoxifen male n = 6.

Tamoxifen exerts estrogen-agonistic effects on NG2 cells and estrogen-antagonist effects on monocytes, stimulating and repressing proliferation, respectively [56–58]. Therefore, it is possible that the number of proliferating cells did not change due to offsetting agonist and antagonist effects. To further scrutinize cell proliferation, we analyzed total and dividing NG2 cells and phagocytic macrophages/microglia in the spared white matter rostral and caudal to the lesion epicenter. Of all the EdU+ cells, ~21–26% were NG2+ and ~25% were CD68+ in control and tamoxifen female mice (Fig 7E and 7I). Similarly, ~22–26% were NG2+ and ~20–25% were CD68+ in control and tamoxifen male mice (Fig 7I). No significant differences associated with sex or drug administration were detected for total cell number or cell proliferation of NG2+ cells or CD68+ macrophages/microglia (Fig 7).

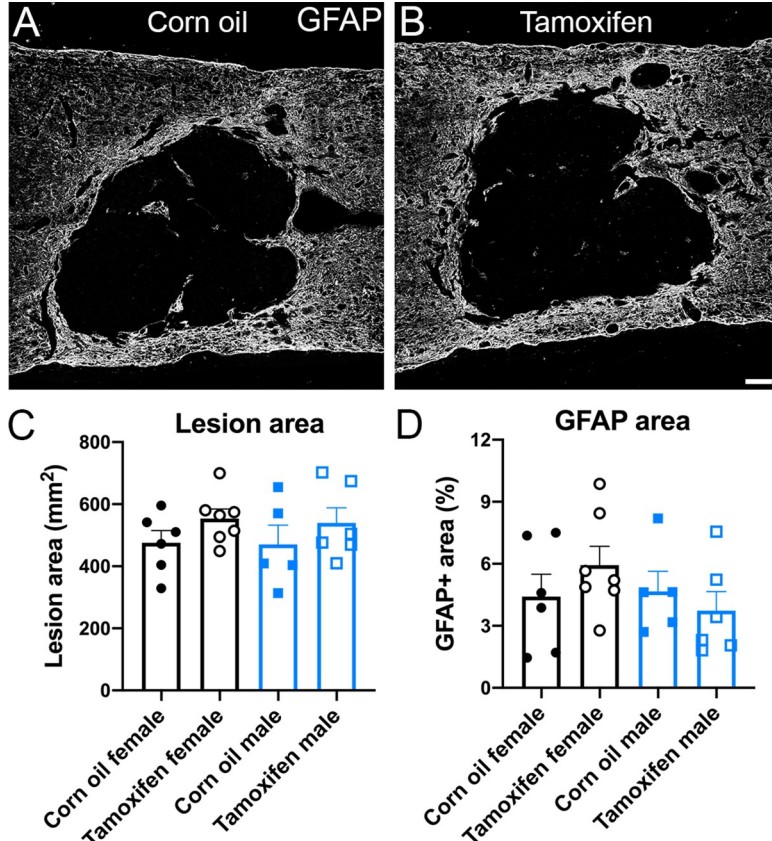

**Fig 6. Astrocyte reactivity and lesion size are not altered by delayed short-term tamoxifen gavage after SCI.** (A-B) Tiled confocal z-stacks of GFAP immunoreactivity in (A) control and (B) tamoxifen mice. Scale bar = 100 μm. Quantification of (C) lesion area and (D) GFAP proportional area revealed no significant differences between groups. Data presented as mean + SEM. One-way ANOVA with Tukey *post hoc* test; corn oil female n = 6; tamoxifen female n = 7; corn oil male n = 5; tamoxifen male n = 6.

## Delayed short-term tamoxifen gavage after SCI does not change total oligodendrocyte numbers, demyelination, or remyelination in the spared white matter

In the ethidium bromide model of demyelination, tamoxifen stimulates OL differentiation and increases the number of remyelinated axons in rats [35]. To determine if tamoxifen had similar neurorestorative effects after SCI, changes in OL number and myelination were examined in the lateral white matter 0.5 mm, 1 mm, and 2 mm rostral and caudal to the injury site.

The total number of OLs was quantified by counting GST$\pi$ and DAPI co-localization. Control female mice averaged 191 ± 55 OLs 0.5 mm rostral and 267 ± 54 OLs 0.5 mm caudal to the lesion, and tamoxifen female mice averaged 370 ± 31 OLs 0.5 mm rostral and 314 ± 21 OLs 0.5 mm caudal to the lesion (Fig 8A and 8B). Control male mice averaged 357 ± 25 OLs 0.5 mm rostral and 272 ± 25 OLs 0.5 mm caudal to the lesion, and tamoxifen male mice averaged 242 ± 29 OLs 0.5 mm rostral and 247 ± 28 OLs 0.5 mm caudal to the lesion (Fig 8B). In both groups, neither sex nor tamoxifen administration lead to any significant differences in the number of OLs at the lesion border or in the distal white matter (Fig 8A and 8B).

To gain insight into the effects of tamoxifen on myelination, we assessed demyelination and remyelination after SCI in control and tamoxifen mice. Since myelin labels poorly with

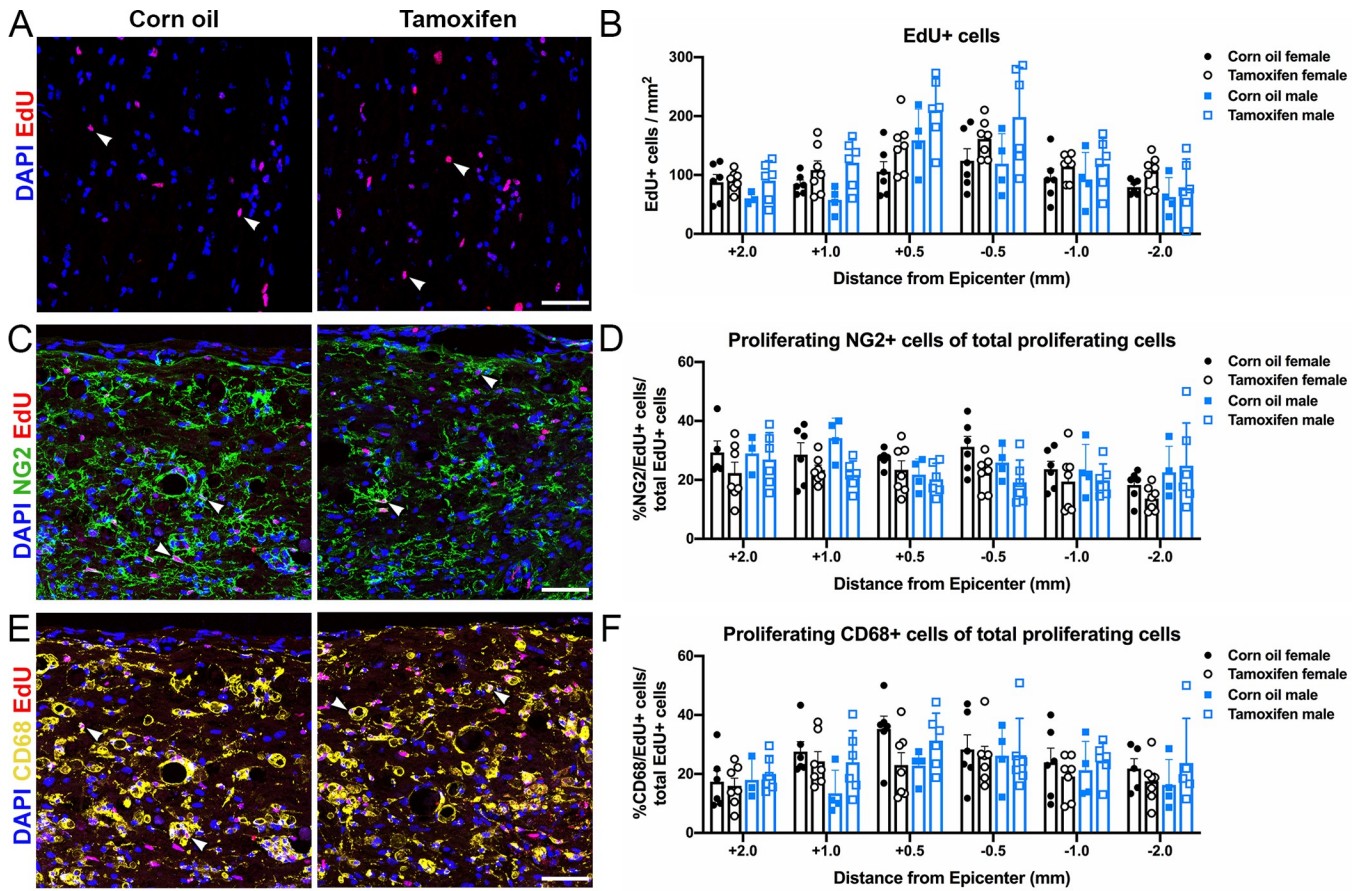

**Fig 7. Delayed short-term tamoxifen treatment does not alter cell proliferation.** (A, C, E) Representative confocal images of (A) proliferating cells (EdU/ DAPI+ cells), (C) proliferating NG2 cells (NG2/EdU/DAPI+ cells), and (E) proliferating phagocytic macrophages/monocytes (CD68/EdU/DAPI+ cells) in the lateral white matter of control and tamoxifen mice. Scale bar = 75 μm. (B) Quantification of the total number of proliferating cells 2 mm, 1mm, and 0.5 mm rostral and caudal to the lesion epicenter. (D, F) Percent of proliferating (D) NG2 cells and (F) phagocytic macrophages/monocytes normalized to total proliferating cells at each sample distance rostral and caudal to the lesion. (B, D, F) The data are not significantly different between control and tamoxifen mice or female and male mice at any distance from the lesion epicenter. Data presented as mean + SEM. Mixed-effects analysis with Šídák *post hoc* test; corn oil female n = 6; tamoxifen female n = 7; corn oil male n = 5; tamoxifen male n = 6.

histology, myelinated and demyelinated axons are often quantified via node of Ranvier labeling [59]. Healthy, myelinated axons contain organized clusters of proteins within and surrounding the node of Ranvier, which includes contactin associated protein (Caspr) in the paranode and the potassium voltage-gated ion channel, Kv1.2, in the juxtaparanode [60–62]. After SCI, axons become demyelinated and lose these structured nodal domains as Caspr and Kv1.2 spread along the axon [61, 63]. Therefore, as a measure of axon myelination, we quantified the number of intact nodes of Ranvier, which is defined as two Caspr profiles flanked by Kv1.2 profiles (Fig 9A and 9B). We then quantified the number of intact nodes per axon profile (Fig 9C). Neither analysis revealed significant differences between control and tamoxifen mice or between female and male mice.

To further examine the structure of myelin, semi-thin epon embedded tissue sections were stained for toluidine blue (Fig 9D). A majority of axons were surrounded by thick myelin sheaths, and in smaller quantities, axons either had thin myelin sheaths, indicative of remyelination, or completely lacked myelin (Fig 9D–9F). The percent of remyelinated, demyelinated, and spared myelinated axons was not different between groups near or distal to the lesion

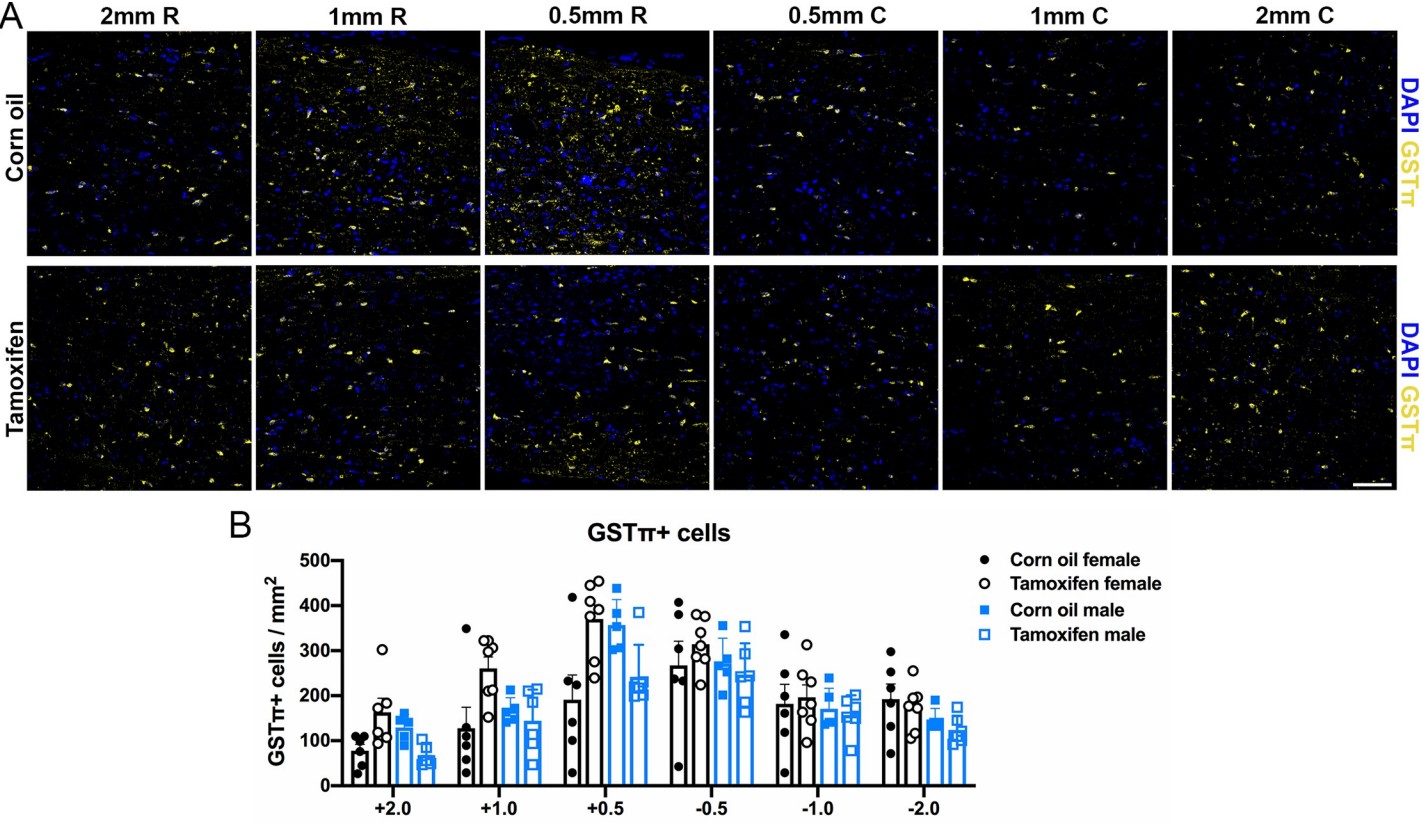

**Fig 8. Oligodendrocyte number does not change after delayed short-term tamoxifen gavage.** (A) Representative confocal images of OLs (GSTπ/DAPI+ cells) in the lateral white matter. Scale bar = 75 μm. (B) Quantification of the total number of OLs 2 mm, 1mm, and 0.5 mm rostral and caudal to the lesion epicenter revealed no significant differences between groups. Data presented as mean + SEM. Two-way repeated measures ANOVA with Šídák *post hoc* test; corn oil female n = 6; tamoxifen female n = 7; corn oil male n = 5; tamoxifen male n = 6.

epicenter (Fig 9F). Collectively, these data show that axon myelination is not changed when tamoxifen is given short-term at the onset of endogenous remyelination.

## Discussion

Giving tamoxifen to transgenic mouse models is necessary to gain temporal control of activating and deactivating genes in specific cells. To determine the potential impact of delayed tamoxifen treatment on motor and anatomical repair after SCI, female and male mice received a spinal cord contusion followed by corn oil or tamoxifen gavage from 19 – 22d post-injury. Our data revealed that all mice experienced similar levels of hindlimb function and tissue repair, regardless of sex or treatment group.

Although tamoxifen transiently increased lethargy, control and tamoxifen mice scored similarly on the BMS scale and had a similar number of foot falls on the automated horizontal ladder. These results contradict other CNS trauma studies that show tamoxifen improves anatomical and motor recovery in female and male SCI rodents [39–41, 55], but this is likely attributed to differences in the timing of dosing. Previous studies began tamoxifen treatment within 24 hours of injury, then gave low doses of tamoxifen chronically via intraperitoneal injection [39, 41] or intrathecal infusion [40, 55]. In our study, tamoxifen treatment was delayed until 19 days post-injury, then gavaged at a high dose for 4 consecutive days. Following

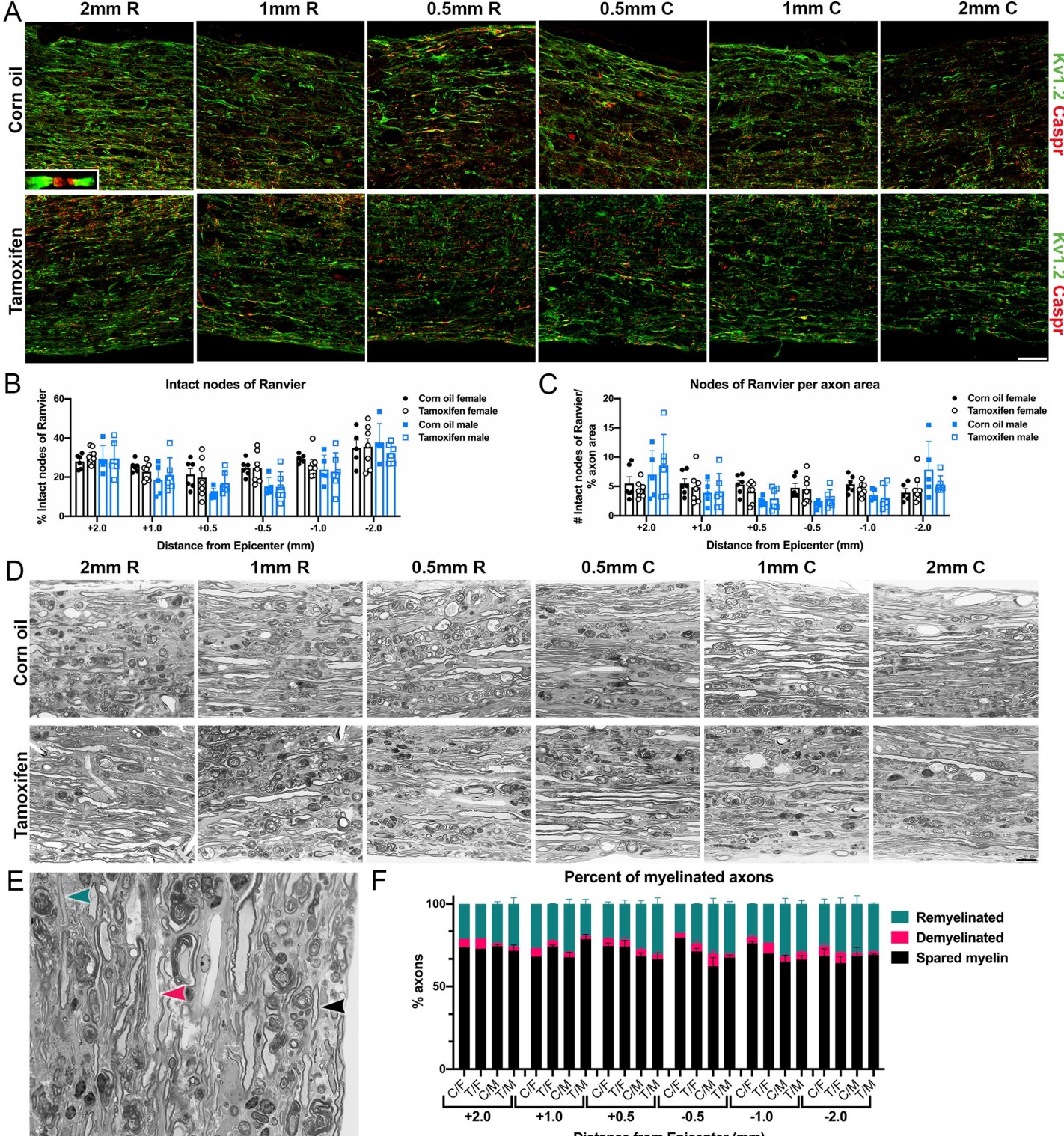

**Fig 9. Demyelination and remyelination are unaffected by delayed short-term tamoxifen administration.** (A) Representative confocal images of longitudinal spinal cords immunolabeled for Kv1.2 and Caspr at 2 mm, 1 mm, and 0.5 mm rostral and caudal to the lesion epicenter. Inset shows a higher-power view of a node of Ranvier. Scale bar = 75 μm. (B) Percent of intact nodes of Ranvier in the lateral white matter at each sampled region. (C) Number of intact nodes of Ranvier normalized to percent axon area (Fig 4) in the lateral white matter at each sampled region. (D) Representative images of semi-thin longitudinal sections stained for toluidine blue from corn oil and tamoxifen mice at 2 mm, 1 mm, and 0.5 mm rostral and caudal to the lesion epicenter. Scale bar = 10 μm. (E) High-power image with arrowheads showing

an example of a spared myelinated axon (black), demyelinated axon (pink), and remyelinated axon (teal). Scale bar = 10 μm. (F) Percent of spared myelinated, demyelinated, and remyelinated axons of total axons in each section. C/F = corn oil female; T/F = tamoxifen female; C/M = corn oil male; T/M = tamoxifen male. (B-C, F) The data are not significantly different between groups at any distance from the lesion epicenter. Data presented as mean + SEM. Mixed-effects analysis with Šídák *post hoc* test. (B-C) Corn oil female n = 6; tamoxifen female n = 7; corn oil male n = 5; tamoxifen male n = 6. (F) Corn oil female n = 1–3; tamoxifen female n = 1–3; corn oil male n = 3; tamoxifen male n = 3.

moderate spinal cord contusion, spontaneous hindlimb motor recovery occurred in all our injury groups until plateauing at 14pi. Thus, tamoxifen was given after spontaneous motor recovery ceased, which may explain a lack of effect of tamoxifen on locomotion.

Immediately after injury the spinal cord undergoes massive cellular and molecular changes, including increased cell death [64, 65], infiltration of inflammation cells in the lesion [2–6], axon degeneration [7–10], and OL and myelin loss [13–16, 66, 67]. Studies that report a neuro-protective effect of tamoxifen have given it during this acute phase, capitalizing on tamoxifen's ability to reduce inflammation, decrease glial reactivity, and promote cell survival [39–42, 55]. Here, we gave tamoxifen 19-22dpi, which is after the glial scar has formed and most white matter, glial, and neuron loss has stabilized [66–70]. Therefore, as expected, tamoxifen did not significantly alter the injury environment. Corn oil- and tamoxifen-gavaged mice had comparable values for lesion size and astrocyte reactivity. In addition, axon sprouting and total axon density near the lesion border and in the lateral white matter rostral and caudal to the lesion cavity was similar between groups. White matter sparing directly correlates with hindlimb motor recovery [71]; therefore, since locomotion was not changed by tamoxifen, it is not surprising that total spared white matter was also unaffected by tamoxifen. Neuronal cell counts were the only significant difference recorded in this study, with tamoxifen mice having fewer neurons overall in the dorsal and ventral horn compare to control mice. This is opposite of previous reports that suggest tamoxifen is neuroprotective and increases neuron survival [39, 41, 42, 55]. Tamoxifen is known to be toxic to humans and rodents at high doses [72–74]; therefore, it is possible that giving tamoxifen at 300 mg/kg, compared to 1 mg/kg [39, 55], was toxic and induced neuronal cell death. However, when examining neurons at specific differences rostral and caudal to the lesion epicenter, tamoxifen did not change neuron density.

Less than 1% of glial cells divide in the uninjured rodent spinal cord [75]; however, after SCI, NG2 cell proliferation increases 50-fold as NG2 cells proliferate to replace NG2 cells and OLs lost from injury [44, 47]. Similarly, phagocytic microglia and infiltrating macrophage proliferation increases 6-fold within the first week of injury compared to uninjured controls [2, 4, 76]. This significant increase in macrophage/microglia and NG2 cell proliferation is maintained for months post-injury [2, 44]; therefore, it is not surprising that our data reveal ~25% of the NG2+ cells and CD68+ cells are EdU+ at 42pi. However, total and proliferating NG2 cells and phagocytic macrophages/microglia numbers were not different between groups. This argues that tamoxifen does not have a lasting effect on cell proliferation.

Injury to the spinal cord results in OL death and demyelination, which impairs neuronal signaling and limits functional recovery [13, 61, 77, 78]. Endogenous NG2 cells then proliferate and differentiate to replace lost OLs and remyelinate spared axons [13, 47, 66, 79, 80]. The Cre/lox system been used in reporter mice to track oligodendrogenesis and myelin synthesis during development [45, 81], after CNS demyelination [82, 83], and after SCI [44, 84, 85]. Since tamoxifen promotes OL differentiation and remyelination [35], it is possible that inducing Cre-recombination with tamoxifen artificially inflates the extent of remyelination. However, our findings show that delayed tamoxifen did not impact myelination as the total number of oligodendrocytes, demyelinated axons, and remyelinated axons were not significantly different between groups. These data indicate that new myelin in reporter mice is formed from endogenous remyelination, and not induced by tamoxifen.

A proposed mechanism for the neuroprotective effects of tamoxifen is cell-specific control of estrogen release. Interestingly, other studies show that female SCI rats recover faster and to a higher level than male SCI rats, which may be due to higher levels of estrogen in females [86, 87]. Estrogen reduces cell death by reducing extracellular glutamate and reactive oxygen species [88–90], and stimulates cell proliferation and differentiation via the extracellular-signal-regulated kinase (ERK) pathway and the AKT/mammalian target of rapamycin (mTOR) pathway [90–93]. Since tamoxifen binds to the estrogen receptor and acts as an agonist in most CNS cells [57, 58], we examined tissue repair and motor recovery in female and male mice. Data analyses revealed no significant differences in histology between female and male mice. Similarly, sex did not impact BMS scores, foot falls on the horizontal ladder, or activity. However, BMS subscores were significantly higher for males at 14dpi. This suggests that male mice gained fine motor control faster than females. Contusion analyses from the Infinite Horizon device revealed no significant differences in force, displacement, or area under curve (data available in data repository); therefore, faster recovery was not due to a less severe injury. The higher subscore at 14dpi is likely because males achieved consistent plantar stepping on the BMS scale, which occurs when mice miss 5 or fewer steps during the 4-minute test. During BMS testing, 5 of 19 female mice experienced a common spasm known as "butt down" during which they lose weight support on their hindlimbs, resulting in missed steps that dropped their BMS subscore; no male mice experienced "butt down." Additional studies are needed to determine if female mice commonly experience these spasms more frequently than male mice. Importantly, this difference cannot be attributed to tamoxifen since male BMS subscores were only higher 14dpi which was before drug treatment. Thus, tamoxifen did not cause lasting sex-specific effects.

Taken together, these data support our hypothesis that delayed short-term tamoxifen gavage does not alter functional recovery, neuroprotection, or neurorestoration after SCI. Therefore, giving tamoxifen to induce Cre-mediated recombination chronically after SCI should not interfere with experimental outcome measures.

## Acknowledgments

The authors greatly acknowledge the excellent technical assistance of Rochelle Deibert and Christina Lepak.

## Author Contributions

**Conceptualization:** Nicole Pukos, Dana M. McTigue.

**Data curation:** Nicole Pukos.

**Formal analysis:** Nicole Pukos.

**Funding acquisition:** Dana M. McTigue.

**Investigation:** Nicole Pukos.

**Methodology:** Nicole Pukos, Dana M. McTigue.

**Project administration:** Dana M. McTigue.

**Resources:** Dana M. McTigue.

**Software:** Nicole Pukos.

**Supervision:** Dana M. McTigue.

**Validation:** Nicole Pukos, Dana M. McTigue.

**Visualization:** Nicole Pukos.

**Writing – original draft:** Nicole Pukos.

**Writing – review & editing:** Nicole Pukos, Dana M. McTigue.

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
