## [Decision Letter · Decision Letter 0]

13 Apr 2020

PONE-D-20-06858

Delayed short-term tamoxifen treatment does not promote remyelination or neuron sparing after spinal cord injury

PLOS ONE

Dear Dr. Mctigue,

Thank you for submitting your manuscript to PLOS ONE. After careful consideration, we feel that it has merit but does not fully meet PLOS ONE’s publication criteria as it currently stands. Therefore, we invite you to submit a revised version of the manuscript that addresses the points raised during the review process.  We have thoroughly reviewed your manuscript and ask you to provide answers only to a couple of points listed by reviewer 1. Although the findings suggest that tamoxifen does not support remyelination and is neither neuroprotective, we believe that such findings are important and advance our knowledge on neuroscience.

We would appreciate receiving your revised manuscript by May 28 2020 11:59PM. To enhance the reproducibility of your results, we recommend that if applicable you deposit your laboratory protocols in protocols.io, where a protocol can be assigned its own identifier (DOI) such that it can be cited independently in the future. For instructions see: http://journals.plos.org/plosone/s/submission-guidelines#loc-laboratory-protocols

We look forward to receiving your revised manuscript.

Kind regards,

Antal Nógrádi, M.D., Ph.D., D.Sc.

Academic Editor

PLOS ONE

Journal Requirements:

2. To comply with PLOS ONE submissions requirements, in your Methods section, please provide additional information on methods of a analgesia.

3. Thank you for including your ethics statement: "All surgical and postoperative care procedures were approved and performed in accordance with the Ohio State University Institutional Animal Care and Use Committee (Protocol Number: 2012A00000030-R2). Mice were anesthetized with a ketamine/xylazine mixture for surgical procedures. A lethal (1.5x) dose of ketamine/xylazine was given for euthanasia, after which mice were transcardially perfused with 4% paraformaldehyde. All efforts were made to minimize animal suffering for the duration of the experiment."

a) Please amend your current ethics statement to confirm that your named ethics committee specifically approved this study.

Reviewers' comments:

Reviewer's Responses to Questions

**Comments to the Author**

1. Is the manuscript technically sound, and do the data support the conclusions?

Reviewer #1: Yes

2. Has the statistical analysis been performed appropriately and rigorously? 

Reviewer #1: Yes

3. Have the authors made all data underlying the findings in their manuscript fully available?

Reviewer #1: Yes

4. Is the manuscript presented in an intelligible fashion and written in standard English?

Reviewer #1: Yes

5. Review Comments to the Author

Reviewer #1: It is well known that immediate administration of tamoxifen, a FDA-approved drug has neuroprotective effect following traumatic injury of spinal cord. This study aimed to identify the effect of delayed short-term tamoxifen treatment on tissue sparing and functional recovery following spinal cord contusion injury in male and female mice. The authors treated mice with tamoxifen for 4 consecutive days at three weeks after injury and examined a number of morphological parameters such as myelination, NG2+ progenitor cell proliferation, astrocyte and macrophage reactivity, neuron and axon numbers and hind limb motor recovery. The authors have made serious efforts to evaluate and explain extensively the details of the morphological and functional results.

However, the delayed short-term tamoxifen administration did not affect the extent of demyelination and remyelination, and did not alter the cell ploriferation and astrogliosis. Based on the present data, delayed tamoxifen treatment has no positive effect on tissue sparing and motor function recovery. Nevertheless, these negative findings are valuable components of spinal cord research.

There are some minor points to be corrected:

1) Fig. 4: Authors quantified the neurons in the gray matter of dorsal and ventral horns rostrally and caudally to the lesion. Was any sprouting observed around the lesion/cavity?

2) Tamoxifen was administrated by oral gavage to spinal cord injured mice. Why was the tamoxifen administrated for 4 days? Please explain.

6. PLOS authors have the option to publish the peer review history of their article (what does this mean?). If published, this will include your full peer review and any attached files.

Reviewer #1: No

---

## [Author Response · Author response to Decision Letter 0]

5 Jun 2020

Reviewer 1

Reviewer comments to the author: It is well known that immediate administration of tamoxifen, a FDA-approved drug has neuroprotective effect following traumatic injury of spinal cord. This study aimed to identify the effect of delayed short-term tamoxifen treatment on tissue sparing and functional recovery following spinal cord contusion injury in male and female mice. The authors treated mice with tamoxifen for 4 consecutive days at three weeks after injury and examined a number of morphological parameters such as myelination, NG2+ progenitor cell proliferation, astrocyte and macrophage reactivity, neuron and axon numbers and hind limb motor recovery. The authors have made serious efforts to evaluate and explain extensively the details of the morphological and functional results. However, the delayed short-term tamoxifen administration did not affect the extent of demyelination and remyelination, and did not alter the cell ploriferation and astrogliosis. Based on the present data, delayed tamoxifen treatment has no positive effect on tissue sparing and motor function recovery. Nevertheless, these negative findings are valuable components of spinal cord research.

Minor issues:

1) Fig. 4: Authors quantified the neurons in the gray matter of dorsal and ventral horns rostrally and caudally to the lesion. Was any sprouting observed around the lesion/cavity?

We have expanded figure 5 to include data on sprouting around the lesion and distal to the lesion using GAP43 immunohistochemistry. No change in sprouting was observed between tamoxifen and control mice, or male and female mice.

2) Tamoxifen was administrated by oral gavage to spinal cord injured mice. Why was the tamoxifen administrated for 4 days? Please explain.

 We agree that the rationale for tamoxifen administration was not clearly explained. In our lab and in the literature, tamoxifen administered via oral gavage for 4 consecutive days is fairly standard and sufficient way to induce Cre-recombination in transgenic mice. The introduction has been updated to include an explanation of this dosing regimen.

---

## [Editor Report · Decision Letter 1]

11 Jun 2020

Delayed short-term tamoxifen treatment does not promote remyelination or neuron sparing after spinal cord injury

PONE-D-20-06858R1

Dear Dr. Mctigue,

We’re pleased to inform you that your manuscript has been judged scientifically suitable for publication and will be formally accepted for publication once it meets all outstanding technical requirements.

Kind regards,

Antal Nógrádi, M.D., Ph.D., D.Sc.

Academic Editor

PLOS ONE
---

## [Editor Report · Acceptance letter]

21 Jul 2020

PONE-D-20-06858R1 

Delayed short-term tamoxifen treatment does not promote remyelination or neuron sparing after spinal cord injury 

Dear Dr. McTigue:

I'm pleased to inform you that your manuscript has been deemed suitable for publication in PLOS ONE. Congratulations! Your manuscript is now with our production department. 

Kind regards, 

on behalf of

Prof. Antal Nógrádi 

Academic Editor

PLOS ONE